# Dormant Season Vegetation Phenology and Eddy Fluxes in Native Tallgrass Prairies of the U.S. Southern Plains

**Pradeep Wagle** [1,*], **Vijaya G. Kakani** [2], **Prasanna H. Gowda** [3], **Xiangming Xiao** [4], **Brian K. Northup** [1], **James P. S. Neel** [1], **Patrick J. Starks** [1], **Jean L. Steiner** [5] and **Stacey A. Gunter** [1]

[1] Grazinglands Research Laboratory, United States of Department of Agriculture-Agricultural Research Service (USDA-ARS), El Reno, OK 73036, USA; brian.northup@usda.gov (B.K.N.); jim.neel@usda.gov (J.P.S.N.); patrick.starks@usda.gov (P.J.S.); stacey.gunter@usda.gov (S.A.G.)

[2] Department of Plant and Soil Sciences, Oklahoma State University, Stillwater, OK 74077, USA; v.g.kakani@okstate.edu

[3] United States of Department of Agriculture-Agricultural Research Service (USDA-ARS) Southeast Area, Stoneville, MS 38776, USA; prasanna.gowda@usda.gov

[4] Department of Microbiology and Plant Biology, Center for Spatial Analysis, University of Oklahoma, Norman, OK 73019, USA; xiangming.xiao@ou.edu

[5] Department of Agronomy, Kansas State University, Manhattan, KS 66506, USA; jlsteiner@ksu.du

* Correspondence: pradeep.wagle@usda.gov

**Abstract:** Carbon dioxide ($CO_2$) fluxes and evapotranspiration (ET) during the non-growing season can contribute significantly to the annual carbon and water budgets of agroecosystems. Comparative studies of vegetation phenology and the dynamics of $CO_2$ fluxes and ET during the dormant season of native tallgrass prairies from different landscape positions under the same climatic regime are scarce. Thus, this study compared the dynamics of satellite-derived vegetation phenology (as captured by the enhanced vegetation index (EVI) and the normalized difference vegetation index (NDVI)) and eddy covariance (EC)-measured $CO_2$ fluxes and ET in six differently managed native tallgrass prairie pastures during dormant seasons (November through March). During December–February, vegetation phenology (EVI and NDVI) and the dynamics of eddy fluxes were comparable across all pastures in most years. Large discrepancies in fluxes were observed during March (the time of the initiation of growth of dominant warm-season grasses) across years and pastures due to the influence of weather conditions and management practices. The results illustrated the interactive effects between prescribed spring burns and rainfall on vegetation phenology (i.e., positive and negative impacts of prescribed spring burns under non-drought and drought conditions, respectively). The EVI better tracked the phenology of tallgrass prairie during the dormant season than did NDVI. Similar EVI and NDVI values for the periods when flux magnitudes were different among pastures and years, most likely due to the satellite sensors' inability to fully observe the presence of some cool-season $C_3$ species under residues, necessitated a multi-level validation approach of using ground-truth observations of species composition, EC measurements, PhenoCam (digital) images, and finer-resolution satellite data to further validate the vegetation phenology derived from the Moderate Resolution Imaging Spectroradiometer (MODIS) during dormant seasons. This study provides novel insights into the dynamics of vegetation phenology, $CO_2$ fluxes, and ET of tallgrass prairie during the dormant season in the U.S. Southern Great Plains.

**Keywords:** $CO_2$ fluxes; drought; evapotranspiration; grazing; vegetation indices

## 1. Introduction

Grasslands are important land uses that support critical ecosystem services and livestock production in every climatic regime and continent of the world [1]. The majority of grasslands have been converted to croplands and urban development worldwide due to expanding human populations [2]. Only less than 1% of the original (pre-European

settlement) > $68 \times 10^6$ ha of tallgrass prairie in the North American Great Plains remains [3]. However, these native prairie pastures are still important for livestock production in many regions of the United States (U.S.).

Management of tallgrass prairie ranges from low input to highly managed approaches based on the needs of ranchers [4,5]. The range of management practices includes prescribed spring burns, different forms of grazing, fertilization, and hay harvest [6,7]. Since the response of prairie grasslands to management practices can be influenced by several factors such as interacting effects with other treatments, climatic conditions, plant communities (i.e., species composition), and landscape positions, inconsistent responses of tallgrass prairies to different management practices have been reported previously [6,8–10].

Eddy covariance (EC) is a widely used technique to measure the exchange of carbon dioxide ($CO_2$) fluxes and evapotranspiration (ET) between ecosystems and the atmosphere [11], on different time scales, ranging from minutes (using wavelet analysis) to hours. In recent years, the number of EC sites has increased globally to measure $CO_2$ fluxes and ET from a diverse range of agroecosystems [12]. Several studies have reported the dynamics of $CO_2$ fluxes and ET in different tallgrass prairies using the EC technique and illustrated that management practices could greatly alter the dynamics of those fluxes during growing seasons [13–18]. Most of these EC studies have focused on the growing season to measure and report the dynamics of $CO_2$ fluxes and ET. Even if some studies report annual carbon and ET budgets, they generally focus on investigating interannual variability of fluxes for growing seasons only and lack detailed investigations of the dynamics of $CO_2$ fluxes and ET during non-growing (i.e., dormant) seasons of perennial ecosystems, including tallgrass prairie. It is also challenging to continuously measure eddy fluxes during winter due to power issues and freezing temperature conditions. However, studies have shown that net ecosystem $CO_2$ exchange (NEE) from various landscapes during non-growing seasons is not trivial and can make significant contributions to the annual carbon budget [19–22]. Similarly, ET during non-growing seasons can contribute to a significant portion of the annual water (i.e., ET) budget. A previous study reported nearly 150 g C m$^{-2}$ of cumulative NEE and 100 mm of cumulative ET loss during the dormant season in a tallgrass prairie [13]. Additionally, understanding the seasonality and magnitudes of fluxes from native prairie over an extended region helps to better assess the impacts on the mesoscale atmospheric environment as the dynamics of the land surface processes and lower atmosphere are tightly coupled [23]. Thus, it is necessary to investigate the variability in $CO_2$ fluxes and ET and to improve our understanding of the key drivers for those variations during non-growing seasons for major land use types.

Satellite remote sensing products have been used for phenology applications due to their ability to capture phenology patterns continuously and retrospectively [24]. However, most studies focused on growing seasons to characterize land surface phenology and investigate the sensitivity of remotely sensed vegetation indices of tallgrass prairie to environmental conditions or management practices [25–27]. As a result, satellite remote sensing products have not been validated satisfactorily for tallgrass prairie during dormant seasons. Based on eddy fluxes and satellite-derived vegetation indices, growing seasons of tallgrass prairie in the U.S. Southern Plains generally begin and end at approximately DOY 100 and 300, respectively [28]. Comparative studies of the vegetation phenology and dynamics of $CO_2$ fluxes and ET using multiple EC systems during the dormant season (i.e., from November to March) in differently managed tallgrass prairies from different landscape positions under the same climatic regime are scarce.

To address this knowledge gap, a long-term integrated Grassland-LivestOck Burning Experiment (iGLOBE) was established with a cluster of six EC systems on differently managed (i.e., different burning and grazing regimes) native tallgrass prairies located in different landscape (e.g., upland, intermediate, and lowland) positions [9]. These pastures are also part of the GRL-FLUXNET (a network of eddy flux towers at the United States Department of Agriculture Agricultural Research Service (USDA-ARS), Grazinglands Research

Laboratory, El Reno, OK, USA) and SP-LTAR (Southern Plains Long-Term Agroecosystem Research) sites.

The major objective of this current study was to compare vegetation phenology (vegetation indices derived from satellite remote sensing observations) and the dynamics of EC-measured $CO_2$ fluxes and ET during dormant seasons (November–March) in multiple native tallgrass prairie pastures. We hypothesized that vegetation phenology, $CO_2$ fluxes, and ET would show no fluctuations during dormant seasons across sites and years even with differences in landscape positions and land/livestock managements. Investigating the dynamics of $CO_2$ fluxes and ET during the dormant season in differently managed tallgrass prairie is critical for improving our understanding of the dynamics of $CO_2$ fluxes and ET in tallgrass prairie to develop resilient forage–livestock systems.

## 2. Materials and Methods

### 2.1. Study Sites

The experimental sites include three different replicate areas of southern native tallgrass prairie at the USDA-ARS, Grazinglands Research Laboratory, El Reno, OK, USA. The dominant grasses present in all the sites are big bluestem (*Andropogon gerardii* Vitman), Indiangrass (*Sorghastrum nutans* (L.) Nash), switchgrass (*Panicum virgatum* L.), and little bluestem (*Schizachryium scoparium* (Michx.) Nash). Dominant soils of the study area include a range of silt loams that are members of the Paleustoll and Argiustoll subgroups [29]. The experimental layout of the studied pastures is shown in Figure S1. Maps showing landscape features and ~500 m Moderate Resolution Imaging Spectroradiometer (MODIS) pixels of each pasture and management details for these pastures were reported in a previous publication [9].

One area (~247 ha) includes four pastures (P13, P14, P15, and P16 of ~60 ha each) that are managed as a single group. These pastures are grazed year-round by 50 cow–calf pairs in a rotation of 30-day grazing periods separated by 90-day rest periods, with ~1.0 t ha$^{-1}$ forage removed annually. These pastures receive prescribed burns in early February on a 4–5-year rotation, with one pasture burned each year. We burned P13 in 2013 and 2018, P14 in 2014 and 2019, P15 in 2012 and 2017, and P16 in 2011 and 2015. The timing of grazing rotations each year is set to begin on 1 May in the pasture burned that year, to take advantage of uniform amounts of high-quality biomass [5]. The landscape positions are intermediate toe through tread for P13, uppermost tread and upper riser for P14, lowest toe through lowland along a stream for P15, and intermediate upper riser to upper toe for P16.

A second pasture (P18 of ~32 ha) is divided into nine paddocks, with eight paddocks (one paddock was an ungrazed control) grazed by yearling stocker cattle in different intensive early stocking treatments for different lengths of time (from 30 to 60 days) during mid-May to mid-July, with the planned level of ~1.5 t ha$^{-1}$ forage removal [9]. This pasture receives an annual prescribed burn in February. The landscape position of this pasture is the toe through the tread.

A third pasture (P20, ~36 ha), which also receives an annual spring burn in February, is harvested for hay (no grazing) in early July, which represents an alternative (uniform) form of intensive defoliation during the early growing. The landscape position of this pasture is the riser through the tread. season.

### 2.2. Eddy Covariance Measurements

The cluster of EC systems was installed in six native prairie pastures to collect $CO_2$, water vapor ($H_2O$), and surface energy fluxes. The EC systems are composed of an open path infrared gas analyzer (LI-7500 RS, LI−COR Inc., Lincoln, NE, USA) and a 3D sonic anemometer (CSAT3, Campbell Scientific Inc., Logan, UT, USA), mounted at 2.5 m tall from the ground surface in all sites. The EC systems were deployed in each field, with enough fetch lengths of several hundred meters in all directions (Figure S1). An Oklahoma Mesonet station, which is a part of more than 110 Mesonet stations across Oklahoma [30], is located in P13, and all pastures were within a few kilometers of the Mesonet station. Rainfall

data for this study were obtained from this El Reno Mesonet station (http://mesonet.org/ (accessed on 9 August 2021)). The 30-year mean values of daily, monthly, and annual air temperature and rainfall were computed for the 1981–2010 period to compare with those for the study periods.

Raw EC data were processed in the EddyPro software (LI-COR Inc., Lincoln, NE, USA), version 7.0.6 in express mode (i.e., default settings), to compute 30 min values of $CO_2$, $H_2O$, and surface energy fluxes. We screened poor-quality data with quality flag 2 (bad-quality fluxes) and unreliable fluxes (beyond the ranges of −50 to 50 µmol m$^{-2}$ s$^{-1}$ for NEE, −200 to 500 W m$^{-2}$ for sensible heat (H), and −200 to 800 W m$^{-2}$ for latent heat (LE)) [31–33]. Additionally, statistical outliers (>3.5 times the standard deviation for every 14 days) were excluded [34].

We used the REddyProc package from the Max Planck Institute for Biogeochemistry, Germany (https://www.bgc-jena.mpg.de/bgi/index.php/Services/REddyProcWebRPackage (accessed on 15 December 2021)) for filling gaps in eddy fluxes and meteorological variables and for partitioning NEE into gross primary production (GPP) and ecosystem respiration (ER) [35]. Gap-filling procedures in the REddyProc included look-up tables, mean diurnal course, and marginal distribution sampling [35]. For NEE partitioning, we employed a nighttime-based flux partitioning method [36] that uses an Arrhenius-type [37] temperature response function of nighttime NEE (=ER). Fluxes for low turbulent periods (u* < 0.2 m s$^{-1}$) were also removed and gap filled in the REddyProc. Diurnal mean values of fluxes were determined using non-gap-filled best-quality fluxes (i.e., the quality flag of 0) only. Cumulative fluxes were derived using gap-filled flux data. Uncertainty estimates (due to gap filling) of cumulative ET and NEE were derived from standard deviations (SD) of the data points used to fill the gaps. A negative NEE indicated a net gain of $CO_2$ by the prairie pastures in this study.

### 2.3. Satellite Remote Sensing Data

The 8-day composite MOD09A1 (~500 m × 500 m) and 16-day composite MOD13Q1 (~250 m × 250 m) data for individual Moderate Resolution Imaging Spectroradiometer (MODIS) pixels that contained EC towers were obtained from the Oak Ridge National Laboratory's Distributed Active Archive Center. The 500 m MODIS pixels included some portions of neighboring pasture(s) in most cases, while the 250 m MODIS pixels were located within the pastures. We used both products in this study due to their different spatial and temporal resolutions. We calculated the enhanced vegetation index (EVI) from blue, red, and near-infrared bands [38] and the normalized difference vegetation index (NDVI) from red and near-infrared bands [39]. Image pixels affected by clouds were excluded from EVI and NDVI computations. Interannual variability in vegetation phenology was compared for six (from 2015–2016 to 2020–2021) dormant seasons in each pasture using EVI and NDVI of both spatial resolutions. Multi-year mean NDVI and EVI values were derived by averaging them for the same day of the year. The 8-day (500 m) NDVI and EVI values had some gaps (i.e., missing values), but the 16-day (250 m) NDVI and EVI values did not have any gaps. Thus, the 16-day composite NDVI and EVI values from the 250 m MODIS pixels were summed or averaged to derive cumulative or average NDVI and EVI values in this study. Standard deviations (SD) were also computed for average or cumulative NDVI and EVI values. We also used GPP fluxes, derived from EC measurements, to further explain discrepancies in vegetation phenology as GPP is a more direct reflection of vegetation greenness.

## 3. Results and Discussion

### 3.1. Weather Conditions for the Study Period

The study area has a temperate continental climate, with a long-term (1981–2010) mean annual daily air temperature ($T_a$) of ~15 °C and mean annual rainfall of ~925 mm [40]. Monthly $T_a$ and rainfall for the study period (2015–2021) are compared with the 30-year means in Figure 1. The 2018–2019 and 2020–2021 non-growing seasons (November through

March of the following year) were cooler (daily average $T_a$ of 4.52 °C and 5.60 °C, respectively), while the 2015–2016 and 2016–2017 non-growing seasons were warmer (daily average $T_a$ of 7.44 °C and 7.85 °C, respectively) as compared to the 30-year mean of 6.2 °C. Temperatures during February–March were warmer in 2016 and 2017 and cooler in 2019 and 2020. February was unusually cooler in 2021 (monthly average $T_a$ of −1.6 °C vs. 30-year mean of 5.3 °C) and March was unusually cooler in 2019 (monthly average $T_a$ of 7.5 °C vs. 30-year mean of 10 °C). The sites received 88, 78, 22, 70, 107, and 63%, respectively, of the 30-year mean rainfall (266 mm) during November–March of 2015–2016, 2016–2017, 2017–2018, 2018–2019, 2019–2020, and 2020–2021. February–March 2017 was the wettest (1.28 times the 30-year mean of 128 mm for the same period), while February–March of 2018 and 2021 received only 38 and 27% of the 30-year mean rainfall for the same period. Overall, the February–March period was both warmest and wettest in 2017, warmer but relatively drier in 2016, and both cooler and drier in 2019 and 2021.

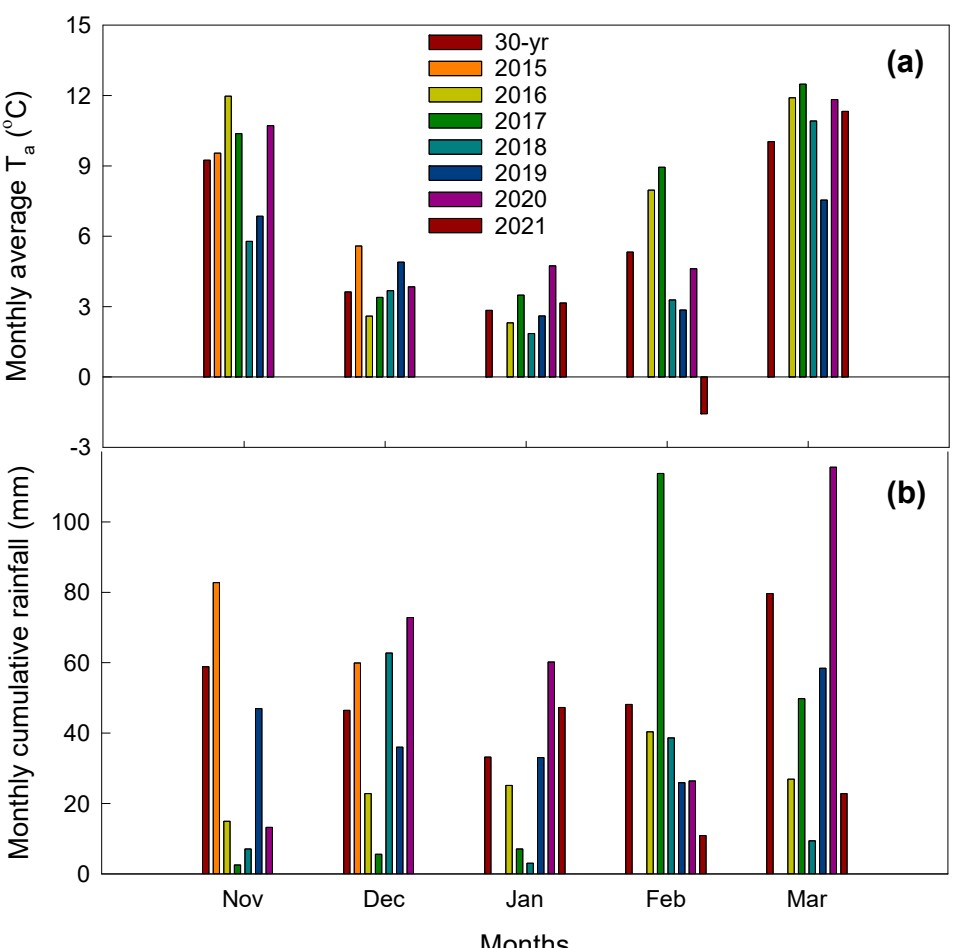

**Figure 1.** Monthly average air temperature ($T_a$) (**a**) and cumulative rainfall (**b**) during the non-growing season (November–March) of native prairie for the study period (2015–2021) as compared to 30-year (1981–2010) means at the El Reno Mesonet station.

When monthly average solar radiation (MJ m$^{-2}$ d$^{-1}$) was compared during dormant seasons for the study period, the range was smaller during November–January across years: 10.1–11.5 in November, 8–9.1 in December, and 9.5–11.8 in January (Figure 2). In contrast, the range was relatively larger during February (from 9.6 in 2019 to 15.3 in 2016) and March (from 15.2 in 2020 to 18.1 in 2016). The sites received the highest solar radiation in 2016 and the lowest in 2019 during February–March.

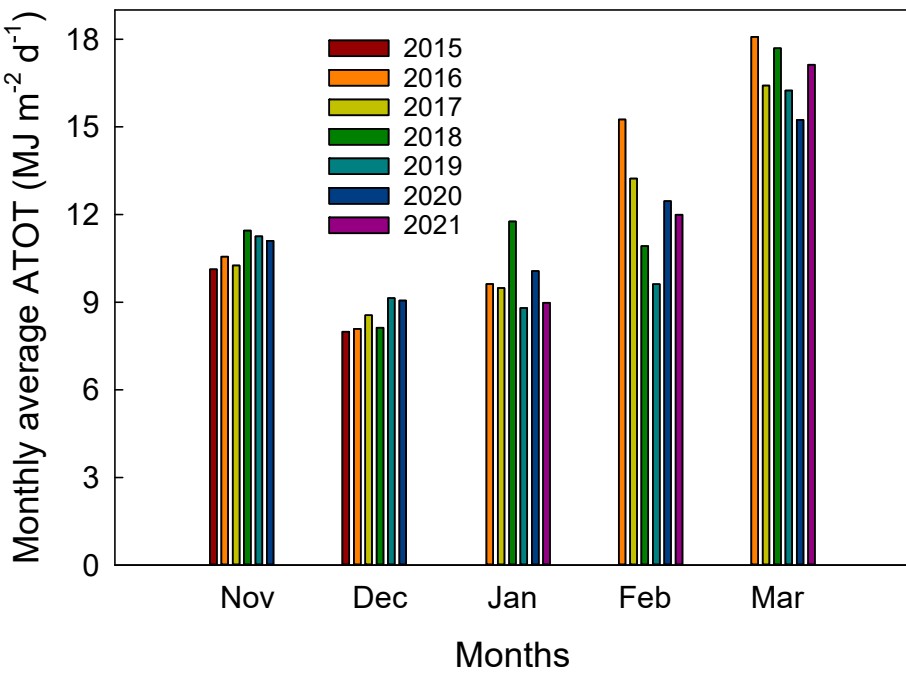

**Figure 2.** Monthly average of daily total solar radiation (ATOT) during the non-growing season (November–March) of native prairie at the El Reno Mesonet station.

### 3.2. Interannual Variability in Vegetation Phenology during Dormant Seasons

When $EVI_{250m}$ and $NDVI_{250m}$ were compared for six (from 2015–2016 to 2020–2021) dormant seasons, the EVI (Figure 3a) showed a larger interannual variability than did NDVI (Figure 3b). There were some differences in the patterns of NDVI and EVI among pastures during the dormant seasons as they were influenced by weather conditions (e.g., rainfall) and management practices (e.g., burning, grazing, or hay harvest). The EVI declined immediately after burning (e.g., P13 in 2018, P14 in 2019, P15 in 2017, and P16 in 2021), but this reduction was not evident from the NDVI. The EVI is an improved form of the NDVI, which employs an additional blue band for atmospheric corrections and a background brightness factor (L) for background noise corrections [38]. The failure of NDVI to show reductions in the vegetation index after burning can be attributed to darker soils (e.g., burned soils), resulting in higher values of vegetation index for a given amount of vegetation if the soil background effect is not removed [41]. These results indicate a greater potential of EVI than NDVI to track vegetation phenology in tallgrass prairie during dormant seasons.

The results illustrated the interactive effects between prescribed spring burns and rainfall on vegetation phenology. The EVI values were lowest until late March of 2018 for P13 and 2021 for P16 than in other years due to drier conditions after burning (Figure 3a). In comparison, EVI and NDVI increased rapidly in P15 after burning in 2017 due to wetter conditions after burning. During 2016 and 2017, when P18 and P20 were burned, EVI and NDVI values increased rapidly due to warmer and wetter conditions than during other years (Figure 3a,b). These results capture the combined positive impacts of the combination of prescribed burns and non-drought conditions on enhanced growth and early greening of vegetation in March. Greater soil temperature and increased solar radiation at the soil surface following burning and enhanced mineralization of soil nitrogen by burning litter layer induce earlier greening of vegetation in the absence of water stress [7,9,42]. However, the results also illustrated the negative impacts of prescribed spring burns on the greening of vegetation during drought conditions due to reduced soil moisture. Soil moisture was substantially reduced by burning at all depths up to 1.5 m in the Bluestem Prairie in Kansas [43].

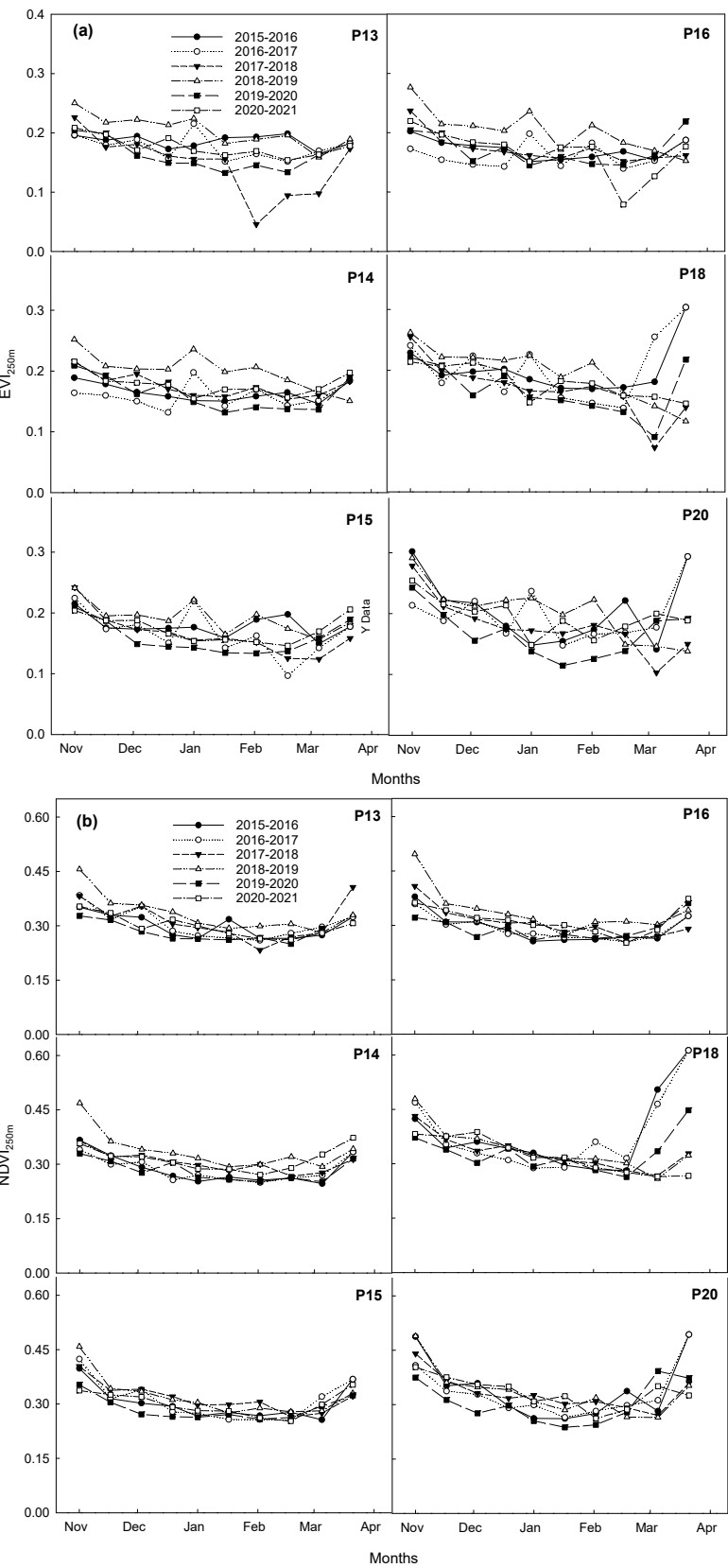

**Figure 3.** Interannual variability of the Moderate Resolution Imaging Spectroradiometer-derived enhanced vegetation index (EVI) (**a**) and normalized difference vegetation index (NDVI) (**b**) at 250 m spatial resolutions during the non-growing season (November–March) of native prairie.

Excluding years when good rainfall (i.e., no occurrence of droughts) occurred in combination with a prescribed burn, there were no observable larger sums of $EVI_{250m}$ and $NDVI_{250m}$ during warmer and wetter dormant seasons compared to cooler and drier dormant seasons in any of P13, P14, P15, and P16 that were burned in 4–5-year rotations (data not shown). Detritus accumulation limits the early greening of vegetation due to decreased solar radiation and soil temperature at the soil surface [7]. In comparison, annually burned P18 and P20 had larger dormant season sums of $EVI_{250m}$ and $NDVI_{250m}$ during warmer and wetter years (e.g., 2016 and 2017) compared to other years, due to the positive impact of the combination of prescribed burns and rainfall. Burning of the litter layer increases soil temperature and solar radiation at the soil surface and induces the early greening of vegetation during non-drought conditions [7,9]. This result was supported by GPP fluxes, which are a more direct reflection of vegetation greenness than ET and NEE fluxes that include the non-stomatal fluxes of evaporation and respiration, respectively. For example, cumulative GPP for 24–31 March of 2016 and 2018 in P18 was approximately 27 and 5 g C m$^{-2}$, respectively. More details on $CO_2$ fluxes are provided in a later section.

The late senescence of newer vegetation growth (i.e., regrowth) in hay harvested (P20) or intensive grazing (P18) pastures during November of good rainfall years (i.e., higher rainfall during October–November) caused some interannual variabilities in vegetation phenology during the dormant season, as illustrated by higher EVI values. In P20, $EVI_{250m}$ values on 1 November were 0.28–0.30 when October–November rainfall sums were higher in 2015 (149 mm), 2017 (111 mm), and 2018 (171 mm), while the $EVI_{250m}$ was 0.21 on 1 November in 2016 (30 mm). However, $EVI_{250m}$ values were similar (0.20–0.22) after mid-November in all years, most likely due to frost damage. Similarly, the $EVI_{250m}$ in P18 was 0.26 on 1 November in 2018 compared to 0.21–0.22 in 2019 and 2020, which was supported by GPP fluxes. Cumulative GPP for 1–10 November in P18 was ~20 g C m$^{-2}$ in 2018 (October–November rainfall of 171 mm) but ~10 g C m$^{-2}$ in 2019 and 2020 (October–November rainfall of 115 and 92 mm, respectively). These results illustrated the positive impacts of rainfall on the regrowth of vegetation, even in the later parts of the growing seasons. In comparison, a previous study on old world bluestem reported a small impact of baling (i.e., hay harvest in summer) on seasonal GPP because increased GPP rates, driven by large rainfall events, in the post-baling period (i.e., fall) offset the reduction in GPP caused by baling [26].

### 3.3. Comparison of Vegetation Phenology during Dormant Season among Pastures

Multi-year (2015–2021) mean EVI and NDVI values for dormant seasons (November–March) were mostly similar across sites (Figure 4). The EVI and NDVI at 250 m resolutions were more similar than those at 500 m across pastures, most likely due to (1) 8-day vs. 16-day composite values (smoother curve for larger averaging windows) and (2) more homogeneity (no mixed pixels) at 250 m resolutions. Average $EVI_{500m}$ and $NDVI_{500m}$ ranges for P13, P14, P15, and P16 were 0.17–0.19 and 0.32–0.33, respectively. In comparison, average $EVI_{500m}$ and $NDVI_{500m}$ were slightly larger in P18 (0.20 ± 0.04 (SD) and 0.36 ± 0.04, respectively) and P20 (0.22 ± 0.04 and 0.38 ± 0.06, respectively). Similarly, $EVI_{250m}$ and $NDVI_{250m}$ ranges for P13, P14, P15, and P16 were 0.17–0.18 and 0.30–0.31, respectively. Average $EVI_{250m}$ and $NDVI_{250m}$ were also slightly larger in P18 and P20 (0.19 ± 0.02 and 0.33–0.35, respectively). The differences in EVI and NDVI were slightly larger during early November and March for P18 and P20 than for the other four pastures. Average ranges for cumulative $EVI_{500m}$ and $NDVI_{500m}$ in P13, P14, P15, and P16 for dormant seasons (November–March from 2015 to 2021) were 3.46–3.74 and 6.29–6.60, respectively. In comparison, cumulative $EVI_{500m}$ and $NDVI_{500m}$ were slightly larger in P18 (3.94 and 7.25, respectively) and P20 (4.47 and 7.56, respectively). Similarly, cumulative $EVI_{250m}$ and $NDVI_{250m}$ ranges were 1.72–1.79 and 3.02–3.10, respectively in P13, P14, P15, and P16. Cumulative $EVI_{250m}$ and $NDVI_{250m}$ were also slightly larger in P18 (1.86 ± 0.16 and 3.47 ± 0.26, respectively) and P20 (1.91 ± 0.15 and 3.29 ± 0.13, respectively). In comparison, EVI and NDVI values during the December–February period were more

comparable (i.e., similar) across all pastures (Figure 4c,d). Average cumulative $EVI_{250m}$ values were 0.98–1.0 in P13, P14, P15, and P16 and 1.07 ± 0.1 in P18 and P20, while average cumulative $NDVI_{250m}$ values were 1.71–1.73 in P13, P14, P15, and P16 and 1.8–1.9 in P18 and P20.

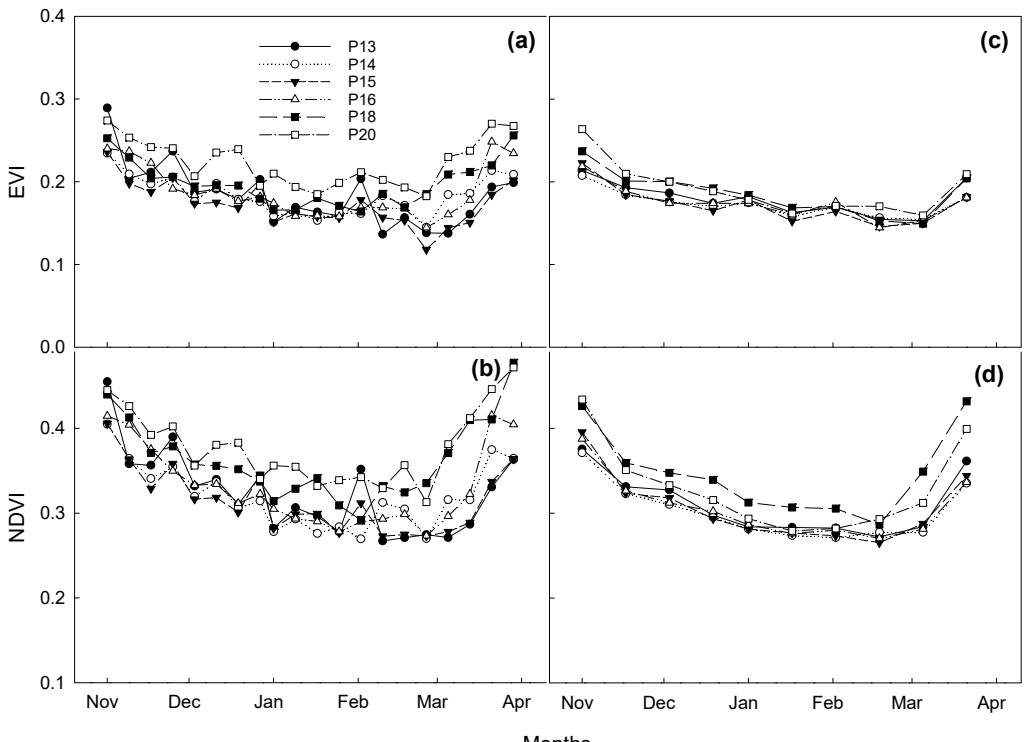

**Figure 4.** Multi-year (2015–2021) Moderate Resolution Imaging Spectroradiometer-derived mean (averaged for the same day of the year) enhanced vegetation index (EVI) and normalized difference vegetation index (NDVI) at 500 m (**a,b**) and 250 m (**c,d**) spatial resolutions for the non-growing season (November–March). Error bars for the means were not plotted to improve the readability of the figure.

Though forms of grazing management are reported to generate little differences in species' composition or plant diversity of prairie ecosystems [4,44], different management practices may impact forage productivity [45]. The P18 was grazed heavily for 2-month periods each year, compared to the lighter year-round rotational grazing in other pastures. The P20 was harvested for hay in early July. Thus, larger values of EVI and NDVI during November in P18 and P20 can be attributed to delayed senescence of newer and more-uniform vegetative growth (regrowth) following intensive grazing (P18) and hay harvesting (P20). For example, average composite (2015–2020) $EVI_{250m}$ values were 0.26 ± 0.03 in P20 (hay harvested in July) and 0.24 ± 0.02 in P18 (heavily grazed from May to July) but were 0.21–0.22 in all other rotationally grazed pastures on 1 November. Similarly, larger values of EVI and NDVI in P18 and P20 during March can be attributed to earlier greening-up due to prescribed spring burns in both pastures compared to pastures (P13, P14, P15, and P16) that were burned on a 4–5-year rotation. For example, the $EVI_{250m}$ values on 21 March in 2016 and 2017 (warmer and wetter periods after prescribed spring burns) in P18 and P20 were ~0.30 but ~0.18 for the same days in the rotational-grazed pastures (P13, P14, P15, and P16). This result was further supported by GPP fluxes. Cumulative GPP for 24–31 March 2016 was 26.5 g C m$^{-2}$ in P18 compared to 8.2 g C m$^{-2}$ in P13 for the same period. However, the $EVI_{250m}$ in the annually burned P18 and P20 was relatively smaller (0.12–0.14) on 21 March when the spring was cooler and drier in 2019 as compared to the $EVI_{250m}$ of 0.15–0.19 on the same day in other pastures because of negative impacts of prescribed burn during drought conditions due to reduced soil moisture. As a result, the monthly

GPP during March 2019 in P18 was 24 g C m$^{-2}$ compared to 26.5 g C m$^{-2}$ in one week (24–31 March) in 2016.

### 3.4. Interannual Variability in CO$_2$ Fluxes and ET during Dormant Seasons

Half-hourly binned diurnal patterns of NEE for a selected pasture (P18), based on the availability of data for a longer period, were comparable from November to February for multiple years, with some larger discrepancies occurring in March (Figure 5a). The monthly average peak NEE during the daytime ranged from −2.2 μmol m$^{-2}$ s$^{-1}$ in December to −3.0 μmol m$^{-2}$ s$^{-1}$ in February. Half-hourly binned diurnal patterns of ET were also comparable from November to February, with the monthly average peak ET during the daytime ranging between 0.03 and 0.04 mm 30-min$^{-1}$ (Figure 5b). The peak daytime ET rates were substantially higher in March 2019 (0.11 mm 30-min$^{-1}$) than in March 2018 (0.05 mm 30-min$^{-1}$) because of greater ET rates after mid-March 2019 due to higher rainfall. Monthly rainfall was 9 mm in March 2018 and 58 mm in March 2019 (52 mm of rainfall from 13 to 30 March). Some discrepancies in diurnal patterns of NEE and ET over multiple years can also be attributed to different averaging periods to create mean monthly diurnal values as we used non-gap-filled half-hourly data to create diurnal means.

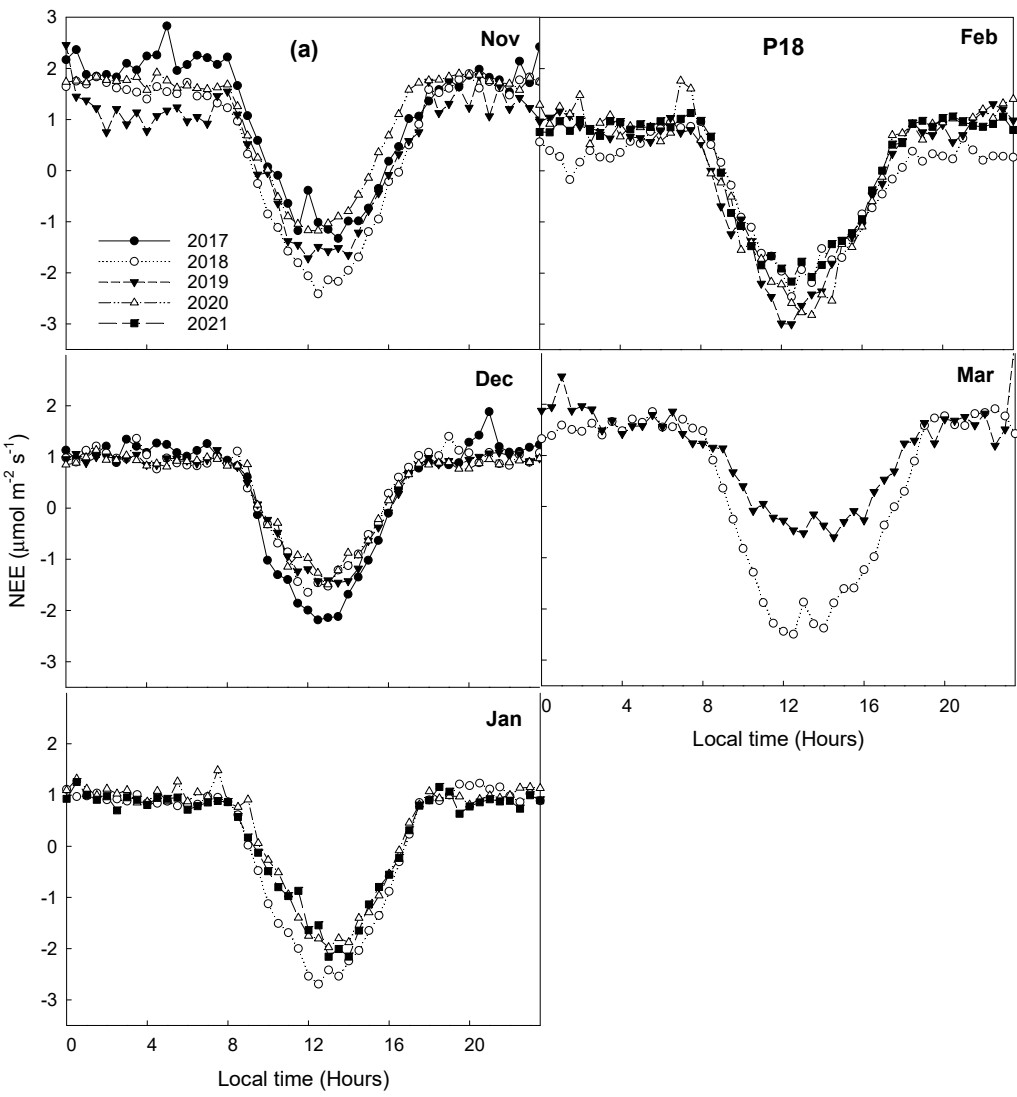

**Figure 5.** *Cont.*

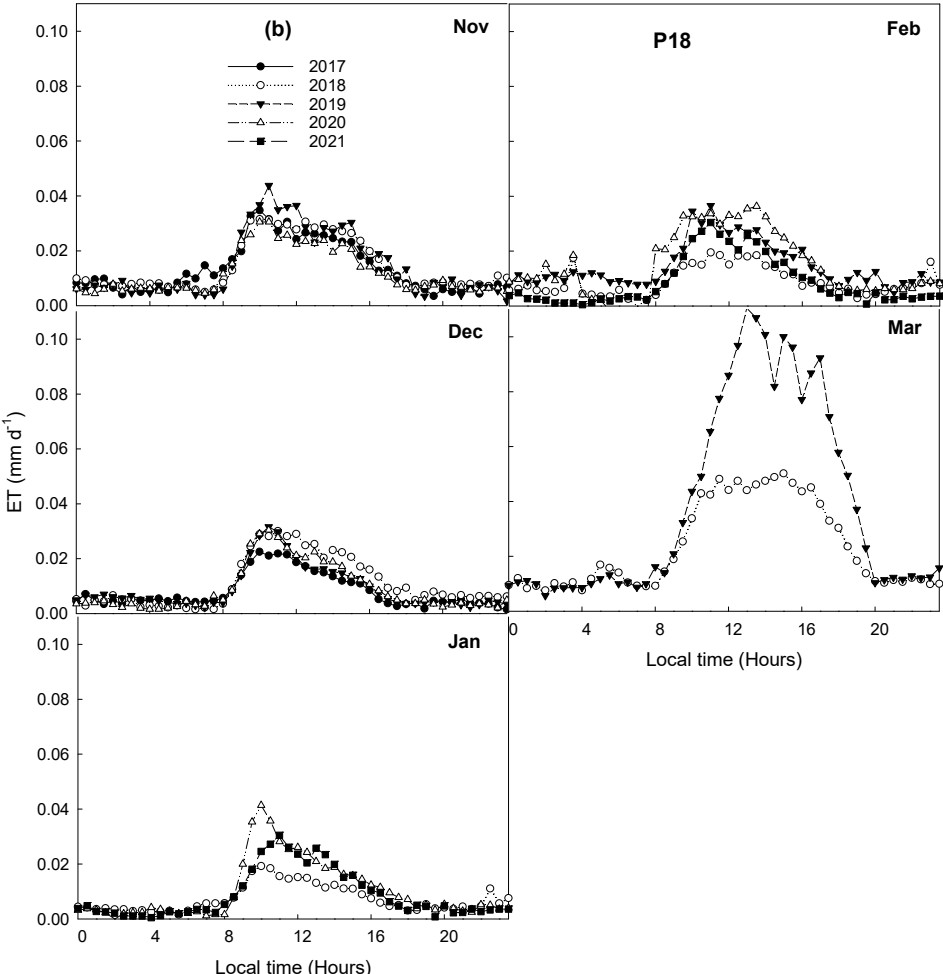

**Figure 5.** Half-hourly binned diurnal patterns of net ecosystem $CO_2$ exchange (NEE) (**a**) and evapotranspiration (ET) (**b**) during non-growing seasons in P18.

As with diurnal patterns, daily ET and NEE patterns in P18 were similar for multiple dormant seasons (Figure 6). Average daily ET from November through February ranged from $0.41 \pm 0.21$ (SD) to $0.54 \pm 0.27$ mm d$^{-1}$, and daily NEE ranged from $0.34 \pm 0.45$ to $0.53 \pm 0.52$ g C m$^{-2}$ d$^{-1}$. Similarly, daily GPP ranged from $0.8 \pm 0.37$ to $1.02 \pm 0.36$ g C m$^{-2}$ d$^{-1}$. Daily rates of $CO_2$ fluxes were also similar in March, though ET rates differed among seasons. For example, average ET rates for DOY 74–90 in P18 were $1.1 \pm 0.4$ (SD) and $2.22 \pm 0.45$ mm d$^{-1}$ in 2018 and 2019, respectively, due to higher rainfall for that period in 2019. However, NEE rates were $0.92 \pm 0.47$ (SD) and $1.05 \pm 0.53$ g C m$^{-2}$ d$^{-1}$ and GPP rates were $0.76 \pm 0.40$ and $0.65 \pm 0.24$ g C m$^{-2}$ d$^{-1}$ for DOY 74–90, 2018 and 2019, respectively.

Evapotranspiration is composed of evaporation ((E) from wet soil, vegetation, and litter) and transpiration ((T) from plant tissues). Evapotranspiration can increase after rainfall due to increased E loss from wet canopies, litters, and soil surfaces, despite the lack of green vegetation to increase T. However, GPP may not increase much after rainfall during the dormant season, as the amounts of green vegetation involved in photosynthesis due to the presence of some $C_3$ grasses and forbs are small in prairie pastures. However, rainfall during the dormant season can have an effect on microbial activities and soil respiration [46]. Rainfall promotes microbial activities by increasing solute mobility and the supply of substrates to decomposers [47]. Studies have reported log–linear relationships between water potential (i.e., a measure of soil water available to microbes) and microbial activities [48,49]. Rainfall events can induce pulses of soil respiration due to the enhanced

microbial activity and physical displacement of $CO_2$-rich soil air by soil water after rainfall events [50].

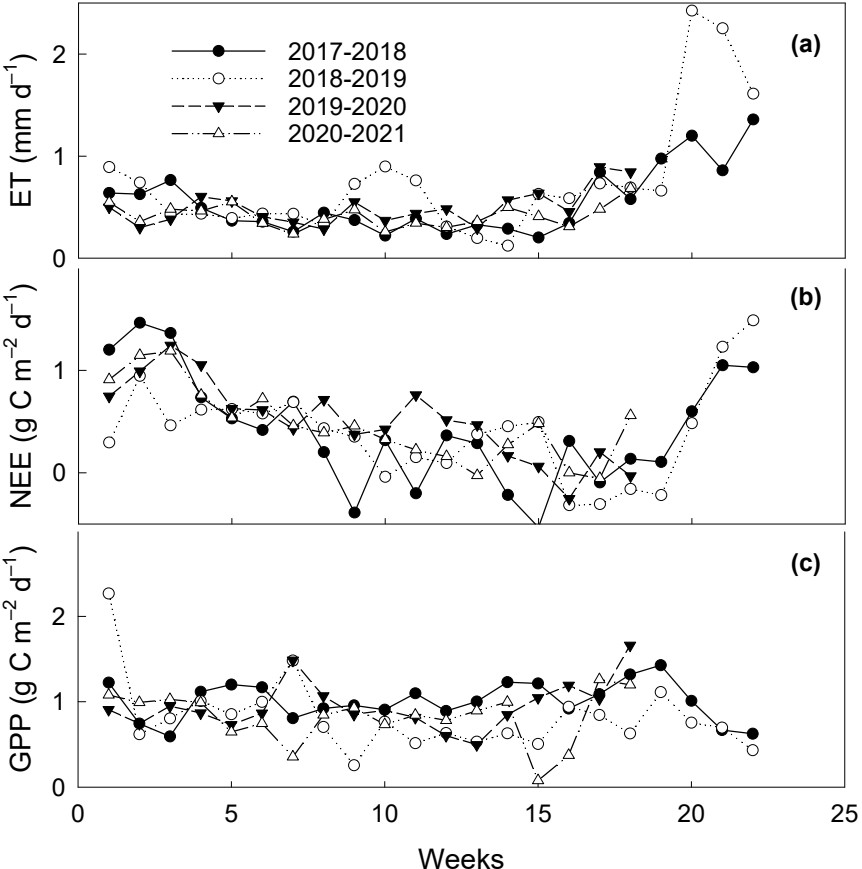

**Figure 6.** Daily patterns of evapotranspiration (ET) (**a**), net ecosystem $CO_2$ exchange (NEE) (**b**), and gross primary production (GPP) (**c**) during non-growing seasons (November–March) in pasture 18.

Monthly NEE, GPP, and ET values were also similar in P18 during the dormant seasons of multiple years (Table 1). The monthly average ET ranged between $12 \pm 1$ mm and $16 \pm 2$ mm from November through February but increased to $39 \pm 13$ (SD) mm in March. The monthly average NEE ranged from 1 g C m$^{-2}$ ($\pm 5$ SD) in February to 28 g C m$^{-2}$ ($\pm 7$ SD) in November. Monthly NEE values were relatively larger (release of $CO_2$) in November (before full senescence of canopy) and March (once roots became more active for the initiation of new aboveground growth). The results indicated that each month during the dormant period was near neutral to a weak source of $CO_2$, consistent with the findings of a previous study [51] from a prairie pasture in Shidler, OK, USA, showing the pasture to be a small source of $CO_2$ (~0.71 g C m$^{-2}$ d$^{-1}$) for the period of mid-November to mid-March. The monthly average GPP ranged between $24 \pm 6$ and $29 \pm 4$ g C m$^{-2}$ for the months between November and March, indicating similar photosynthetic activities during all months of the dormant season. For the November–February period, cumulative ET ranged from $49 \pm 0.34$ mm (2020–2021) to $65 \pm 0.73$ mm (2018–2019) and cumulative NEE ranged from $41 \pm 1.4$ g C m$^{-2}$ (2018–2019) to $64 \pm 2.08$ g C m$^{-2}$ (2019–2020). Similarly, cumulative GPP ranged from 96 g C m$^{-2}$ (2020–2021) to 122 g C m$^{-2}$ (2017–2018). Slightly smaller cumulative ET and GPP values during the 2020–2021 non-growing season could be attributed to drier and unusually cooler conditions in February 2021.

**Table 1.** Monthly evapotranspiration (ET, mm), net ecosystem $CO_2$ exchange (NEE, g C m$^{-2}$), and gross primary production (GPP, g C m$^{-2}$) from November through March in P18. Uncertainty estimates (due to gap filling) of cumulative ET and NEE ($\pm$ SD, standard deviation) were derived from the SD of data points used to fill gaps.

|  | ET | NEE | GPP |
|---|---|---|---|
| November | | | |
| 2017 | 18 ± 0.4 | 34 ± 1.34 | 28 |
| 2018 | 18 ± 0.32 | 18 ± 0.81 | 35 |
| 2019 | 15 ± 0.53 | 31 ± 1.32 | 25 |
| 2020 | 14 ± 0.34 | 30 ± 0.89 | 29 |
| Average | 16 | 28 | 29 |
| SD | 2 | 7 | 4 |
| December | | | |
| 2017 | 11 ± 0.24 | 11 ± 0.61 | 30 |
| 2018 | 14 ± 0.35 | 16 ± 0.8 | 28 |
| 2019 | 13 ± 0.43 | 16 ± 0.9 | 32 |
| 2020 | 12 ± 0.35 | 16 ± 1.1 | 22 |
| Average | 12 | 15 | 28 |
| SD | 1 | 3 | 5 |
| January | | | |
| 2018 | 9 ± 0.31 | 4 ± 0.68 | 31 |
| 2019 | 17 ± 0.37 | 6 ± 0.54 | 18 |
| 2020 | 12 ± 0.47 | 16 ± 0.84 | 22 |
| 2021 | 11 ± 0.35 | 5 ± 0.85 | 26 |
| Average | 12 | 8 | 24 |
| SD | 3 | 5 | 6 |
| February | | | |
| 2018 | 12 ± 0.4 | −6 ± 0.99 | 33 |
| 2019 | 15 ± 0.4 | 2 ± 0.59 | 20 |
| 2020 | 19 ± 0.46 | 1 ± 1.04 | 31 |
| 2021 | 12 ± 0.34 | 6 ± 0.76 | 19 |
| Average | 15 | 1 | 26 |
| SD | 3 | 5 | 7 |
| March | | | |
| 2018 | 30 ± 0.48 | 19 ± 0.95 | 31 |
| 2019 | 48 ± 0.79 | 15 ± 1 | 24 |
| 2020 | | | |
| 2021 | | | |
| Average | 39 | 17 | 27 |
| SD | 13 | 3 | 5 |

Multi-year (2017/18–2020/21) averages of cumulative ET, NEE, and GPP in P18 for the November–February period were 56 ± 7 mm, 52 ± 11 g C m$^{-2}$, and 107 ± 11 g C m$^{-2}$, respectively. During the entire dormant season (November–March), cumulative ET was 81 ± 0.84 mm in 2017–2018 and 112 ± 1.1 mm in 2018–2019, with large differences in ET occurring after mid-March, with cumulative ET of 19 ± 0.42 and 38 ± 0.7 mm for DOY 74–90 in 2018 and 2019, respectively. However, cumulative GPP for the same period of both years was similar (~12 g C m$^{-2}$). Consequently, cumulative $CO_2$ fluxes for the November–March period were similar. Cumulative NEE was 62 ± 2.13 and 57 ± 1.72 g C m$^{-2}$ and cumulative GPP was 153 and 125 g C m$^{-2}$ in 2017–2018 and 2018–2019, respectively. These results illustrated different impacts of rainfall on $CO_2$ fluxes and ET during the dormant season by altering the proportions of stomatal and non-stomatal fluxes, mainly in March with rising temperatures.

When fluxes in P15 were compared for 2 years, cumulative fluxes for the November–February period were comparable in 2019–2020 and 2020–2021 ($63 \pm 1.05$ and $85 \pm 1.3$ mm of ET, $109 \pm 3.31$ and $109 \pm 3.1$ g C m$^{-2}$ of NEE, and 94 and 109 g C m$^{-2}$ of GPP, respectively). In comparison, cumulative ET, NEE, and GPP for the entire dormant season (November–March) were $104 \pm 1.3$ and $117 \pm 1.5$ mm, $148 \pm 3.73$ and $136 \pm 3.7$ g C m$^{-2}$, and 129 and 181 g C m$^{-2}$, respectively, during 2019–2020 and 2020–2021. A larger difference for GPP was noted in March 2021 (72 g C m$^{-2}$) than in March 2020 (35 g C m$^{-2}$) due to greater rates of GPP after mid-March 2021 (cumulative GPP of ~22 vs. 46 g C m$^{-2}$ for DOY 74–90 in 2020 and 2021, respectively).

When two selected non-growing seasons in P13 (2015–2016, 2020–2021) were compared, we observed similarities in cumulative NEE ($100 \pm 2.2$ and $125 \pm 2.3$ g C m$^{-2}$) and ET ($100 \pm 1.4$ and $98 \pm 1.2$ mm) but a larger difference in cumulative GPP (82 and 182 g C m$^{-2}$). These differences were related to larger discrepancies in GPP rates during March. These results from multiple pastures illustrated that interannual variability in cumulative CO$_2$ fluxes and ET during the November–March period was mainly driven by differences in fluxes during March, which is the time of the initiation of the growth of dominant warm-season grasses. However, a combination of drivers can influence the onset of spring growth of warm-season grasses in prairie ecosystems; increasing temperatures and day lengths are likely important factors. Monthly average (30-year mean) temperatures rose over 10 °C in March from ~5 °C in February for the study area.

### 3.5. Comparison of CO$_2$ Fluxes and ET during Dormant Seasons among Pastures

Half-hourly binned diurnal patterns of NEE during the 2020–2021 non-growing season were comparable among pastures (Figure 7a). Peak values of diurnal NEE were also similar (approximately $-2$ µmol m$^{-2}$ s$^{-1}$) across months from November to February, with slightly larger discrepancies recorded in March. Similarly, half-hourly binned diurnal patterns of ET were comparable across multiple months and pastures, with diurnal peak values of ET < 0.05 mm 30-min$^{-1}$ (Figure 7b). Some discrepancies in diurnal patterns of NEE and ET might have occurred due to the different averaging periods used to create monthly diurnal values as we binned non-gap-filled half-hourly data.

Similar to diurnal patterns, daily ET, NEE, and GPP patterns were comparable among pastures (Figure 8). Average daily ET during November–February ranged from $0.42 \pm 0.12$ (SD) to $0.71 \pm 0.25$ mm d$^{-1}$, while daily NEE ranged from $0.81 \pm 0.31$ to $0.96 \pm 0.39$ g C m$^{-2}$ d$^{-1}$ (except smaller average daily NEE of $0.47 \pm 0.37$ g C m$^{-2}$ d$^{-1}$ in P18) and daily GPP ranged from $0.63 \pm 0.28$ to $0.94 \pm 0.55$ g C m$^{-2}$ d$^{-1}$ (except larger average daily GPP of $1.99 \pm 0.91$ g C m$^{-2}$ d$^{-1}$ in P14).

Cumulative ET in P13, P14, P15, and P18, respectively, during November–February of 2020–2021 was $67 \pm 0.95$ (SD), $54 \pm 1.1$, $85 \pm 1.3$, and $48 \pm 0.34$ mm, while cumulative NEE was $105 \pm 2.1$, $100 \pm 3.2$, $109 \pm 3.1$, and $56 \pm 1.82$ g C m$^{-2}$ and cumulative GPP was 195, 229, 109, and 96 g C m$^{-2}$, respectively. Cumulative values of ET, NEE, and GPP were smaller for P18 compared to other pastures. For the November–March period (2020–2021), cumulative ET was $98 \pm 1.2$, $85 \pm 1.14$, and $117 \pm 1.5$ mm, NEE was $125 \pm 2.3$, $115 \pm 3.5$, and $136 \pm 3.7$ g C m$^{-2}$, and GPP was 276, 302, and 181 g C m$^{-2}$ in P13, P14, and P15, respectively. Flux data for some winter months of 2020–2021 were missing for P16 and P20. In the 2019–2020 non-growing season, cumulative ET, NEE, and GPP in P16 were $69 \pm 1.2$ mm, $80 \pm 2.4$ g C m$^{-2}$, and 165 g C m$^{-2}$ for the November–February period and $110 \pm 1.4$ mm, $87 \pm 3.16$ g C m$^{-2}$, and 238 g C m$^{-2}$ for the November–March period, respectively. Similarly, cumulative ET, NEE, and GPP in P20 were $83 \pm 1.21$ mm, $114 \pm 4$ g C m$^{-2}$, and 76 g C m$^{-2}$, respectively, for the November–February period of 2019–2020.

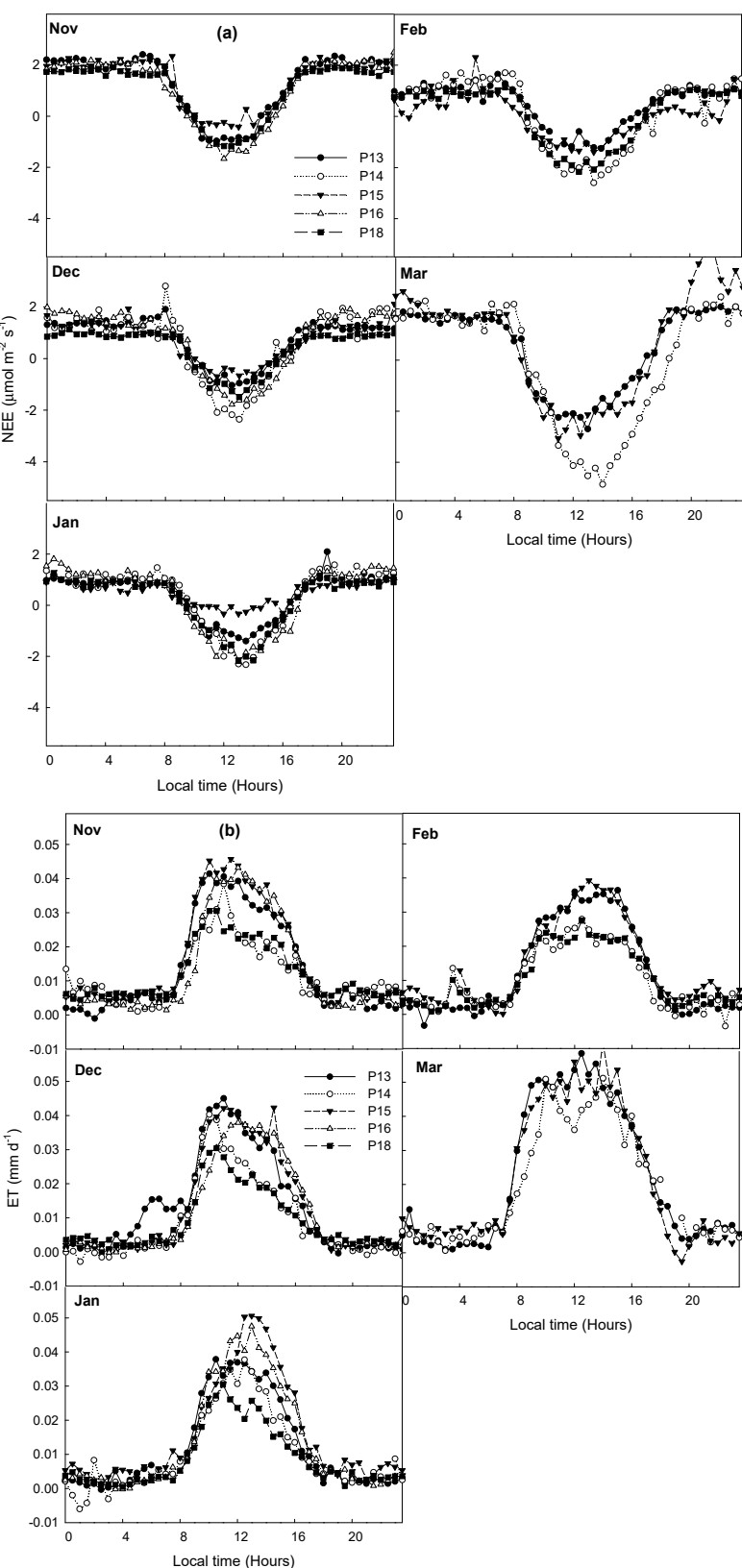

**Figure 7.** Half-hourly binned diurnal patterns of net ecosystem $CO_2$ exchange (NEE) (**a**) and evapo-transpiration (ET) (**b**) in multiple prairie pastures during the 2020–2021 non-growing season.

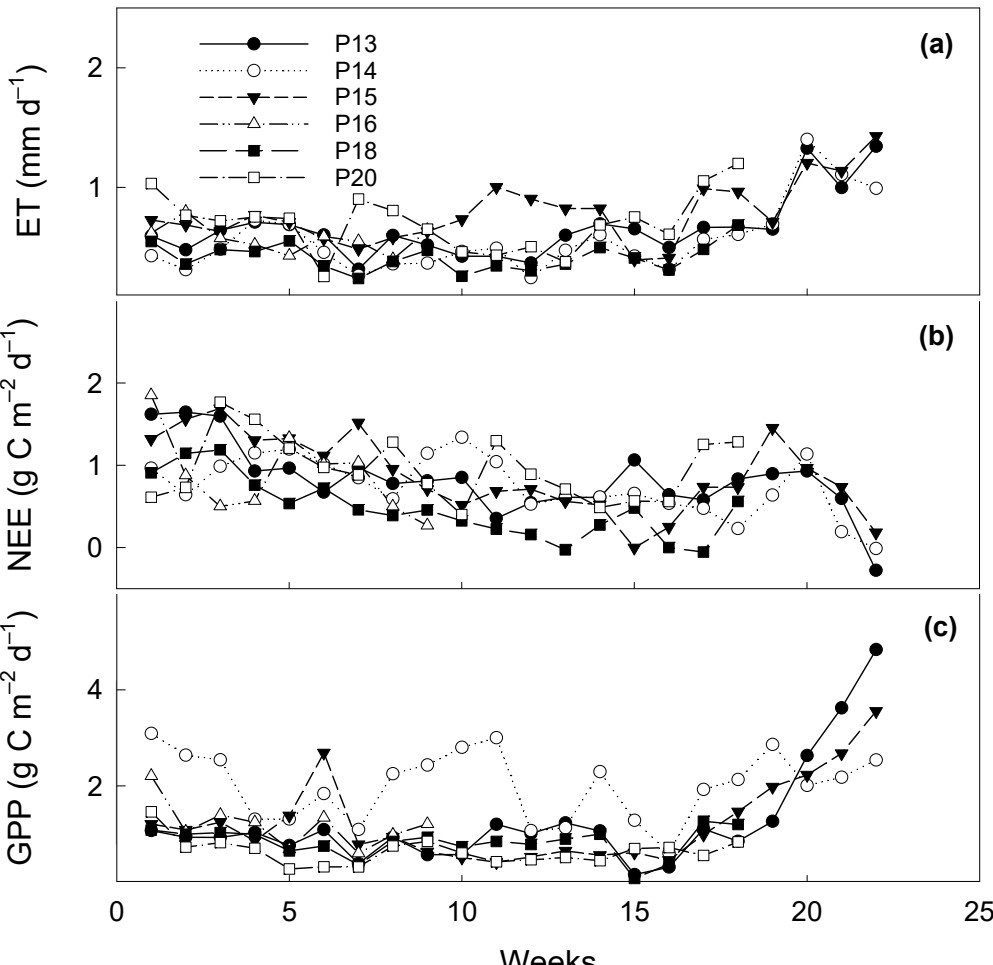

**Figure 8.** Daily patterns of evapotranspiration (ET) (**a**), net ecosystem $CO_2$ exchange (NEE) (**b**), and gross primary production (GPP) (**c**) in multiple prairie pastures during the 2020–2021 non-growing season (November–March), except during the 2019–2020 growing season for P20.

Although relatively larger discrepancies among pastures and years were observed in March, there were some differences in cumulative fluxes among pastures even during the November–February period of the same year. Some differences in vegetation phenology and eddy fluxes among pastures during the dormant season can be attributed to additional factors such as differences in plant communities (i.e., $C_3$:$C_4$ proportions) and landscape positions [52]. As such, the smaller cumulative $CO_2$ fluxes and ET in P18 than in other pastures during the drier (167 mm of rainfall compared to 266 mm for the 30-year mean) 2020–2021 non-growing season can be partially attributed to differences in landscape positions and management practices.

Pasture 18 is located in an upper landscape position and burned annually, P13 and P16 contain a mixture of lowland through upland elements, P14 is largely an upland site, and P15 represents a lowland position. Pastures 13, 14, 15, and P16 are burned in a 4–5-year rotation. In addition, P13, P14, P15, and P16 are rotationally grazed year-round, while P18 is grazed during May through early July only, which results in larger amounts of standing biomass present during winter compared to other pastures.

Frequent burning increases the fraction of $C_4$ grasses and decreases plant species richness and diversity of tallgrass prairie, while less-frequent burning and grazing increase $C_3$ grasses and forbs [53]. The warm-season $C_4$ grasses of prairie ecosystems are dormant during the winter, with only decadent and senesced materials present aboveground. However, cool-season $C_3$ species are metabolically active at low temperatures during winter [54]. Consequently, smaller rates and budgets of $CO_2$ fluxes and ET can be expected during

the dormant season for pastures that are not extensively grazed (P18) and receive annual burning. Furthermore, cumulative $CO_2$ fluxes and ET in P18 showed smaller interannual variations than in other pastures for the November–February period (Table 1), most likely due to the reduced presence of $C_3$ species.

Although we hypothesized that $CO_2$ fluxes and ET would be similar during dormant seasons across differently managed prairie pastures and years, we observed considerable discrepancies in rates and budgets of $CO_2$ fluxes and ET among pastures and years. The NEE is a complicated process since it includes ecosystem respiration (both autotrophic and heterotrophic). Several sub-surface and surface processes such as decomposition of soil organic matter, soil microbial abundance and community, and meteorological conditions (e.g., temperature, moisture, radiation, wind speed) can influence NEE during dormant seasons [22]. Consequently, ET and GPP are more directly influenced by rainfall and the presence of green vegetation, respectively, than NEE. However, we noticed the MODIS-derived EVI and NDVI values were similar for some periods when flux magnitudes were different among pastures and years, indicating MODIS-derived vegetation indices were not fully capable to track the presence of some cool-season $C_3$ species under residues and litter layers of tallgrass prairie. The responses of native prairie systems to different management and environmental factors are varied and complex due to differences in plant community composition (i.e., different proportions of $C_3$ and $C_4$ grasses and $C_3$ forbs) and positions (i.e., slope position, exposure) within landscapes [44,55,56]. There can be interactive impacts of different management and environmental factors on the proportion of $C_3$ species, microbial activities, and soil fluxes.

Documenting changes in species' composition (e.g., the abundance of $C_3$ species) during winter (dormant season) is necessary to explain some discrepancies in dormant season fluxes. A multi-level validation approach of using ground-truth observations of species' composition, EC measurements, PhenoCams (digital cameras), and finer-resolution satellite data is required to further validate the vegetation phenology derived from the MODIS during the dormant season. Such datasets will also allow an investigation of the multifaceted associations (i.e., main and interactive impacts) between vegetation phenology, different management practices, and climatic drivers. Further, several years of EC measurements are needed over the complete years from multiple prairie pastures to more accurately quantify interannual variability in $CO_2$ fluxes and ET in response to controlling factors during both growing and non-growing seasons of this important ecosystem.

## 4. Conclusions

We compared vegetation phenology (vegetation indices derived from satellite remote sensing) and the dynamics of eddy covariance (EC)-measured $CO_2$ fluxes and ET during non-growing seasons (November–March) in six co-located native tallgrass prairie pastures. These pastures were managed under different burn intervals and grazing rotations and are situated in different landscape positions. Although relatively large discrepancies in vegetation phenology and eddy fluxes were observed among pastures and years during March (the time of initiation of growth of dominant warm-season $C_4$ grasses), there were also some noticeable differences in the rates and budgets of eddy fluxes among pastures even during the November–February period of the same year. The MODIS-derived vegetation indices were not fully capable of tracking those differences, most likely due to their inability to capture the presence of some cool-season $C_3$ species, which are metabolically active at low temperatures during winter, under residues and litters of tallgrass prairie. Thus, documentation of the change in species' composition (e.g., the abundance of $C_3$ species) during winter (dormant season) is necessary to explain the discrepancies in fluxes and satisfactorily validate satellite remote sensing vegetation indices for tallgrass prairie in winter. The findings of this study provide a better understanding of the dynamics of vegetation phenology and eddy fluxes in the tallgrass prairie during the dormant season. The collection of multiple years of EC data across multiple pastures and investigating the interactive effects among management practices and climatic conditions on eddy fluxes and

vegetation phenology will further improve our understanding of prairie ecosystems during the dormant season. Such datasets will also aid in deriving quantitative relationships of $CO_2$ fluxes and ET with biophysical variables and improve our understanding of regional carbon and hydrologic cycles and their impacts on the mesoscale atmospheric environment.

**Supplementary Materials:** The following supporting information can be downloaded at: https://www.mdpi.com/article/10.3390/rs14112620/s1. Figure S1. The experimental layout of the studied pastures. Blue dots represent the location of eddy covariance towers.

**Author Contributions:** Formal analysis, P.W.; Funding acquisition, P.W. and S.A.G.; Investigation, P.W.; Methodology, P.W.; Resources, P.W., V.G.K., P.H.G. and X.X.; Writing—original draft, P.W.; Writing—review & editing, V.G.K., P.H.G., X.X., B.K.N., J.P.S.N., P.J.S., J.L.S. and S.A.G. All authors have read and agreed to the published version of the manuscript.

**Funding:** This research was supported in part by the USDA-ARS Office of National Programs (Project number: 3070-21610-003-00D) and USDA-LTAR (Long-Term Agroecosystem Research) Network.

**Acknowledgments:** We greatly acknowledge and appreciate the tremendous support of several USDA-ARS staff members for managing the pastures and assisting in data collection.

**Conflicts of Interest:** The authors declare no conflict of interest for this study.

**Disclaimer:** The mention of trade names or commercial products in this publication is solely for the purpose of providing specific information and does not imply recommendation or endorsement by the U.S. Department of Agriculture. USDA is an equal opportunity provider and employer.

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
