# Peer review of "Dormant Season Vegetation Phenology and Eddy Fluxes in Native Tallgrass Prairies of the U.S. Southern Plains"

_remotesensing, doi:10.3390/rs14112620_

Round 1
Reviewer 1 Report
Dear Authors,
Thank you for your manuscript on the detection of the US Southern Plains CO2 flux based on eddy covariance, an essential atmospheric measurement technique, in relation to satellite remote sensing data.
In my opinion, your manuscript fulfils all aspects,
precise presentation of quantities plus uncertainty statement,
readable diagrams, systematically presented tables, good conclusion.
I have no further remarks or comments. I recommend your manuscript for publication in present form.
Author Response
Thank you so much for your time to review our paper and positive feedback on the manuscript.
Reviewer 2 Report
The subject of this paper falls within the scope of Remote Sensing. It is in general well written with good conclusions, but does not present the applied methods and results comprehensively, I would recommend the manuscript for publication but after major revisions:
General comments
The authors mention LSP when only two vegetation indices are used, without considering phenological dates to, for example, establish differences between the duration of the dormant season each year. It would great to include the use of this phenology or change the name to the behavior of the vegetation indices so as not to mislead.
The material and methods section needs improvement. A description of the data used is given but not of the methodology itself, it is not said what is going to be done to obtain results, what is compared and how, etc. The reader has to interpret what has been done from the results directly.
Section 3.1 is more descriptive of the area than actual results. It should be included in the description of the study area.
The results and discussion section contains parts that should be in material & methods and the results are shown but little discussed until the end of the section (see specific comments).
The references are not homogenized.
Specific comments
Lines 115 – 117: Although they are in previous study, the reader should not have to look for it. A map showing, at least, the location and a short description should be included.
Line 133: Same as the previous case. It could be included in the same figure.
Line 182 / Figure 1: With grayscale you cannot distinguish, for example, 2020 from the 30-year average. It would be advisable to put color in the graph.
Line 192 / Figure 2: Same as Figure 1.
Lines 196 – 197: These things should be included in methodology.
Lines 237 and so on: Results are compared with GPP found in a later section. Refer to it.
Lines 278 – 291: This description is made shortly in the Material and methods section. If it were included in its entirety in that section, this text would not be needed here.
Line 308 / Figure 4: Why the distinction between 500 and 250 m? this is not described and present important differences that can be seen.
Lines 393 – 411: Why are the data for P13 and P15 not included? I suppose that P18 is chosen for its interest, but it should be better explained in the methodology what is going to be done and add graphs or tables of the other plots that are studied.
Lines 447 – 500: Making this discussion more "spread" throughout the section would help to better assimilate the results. If the authors prefer to do it continuously, it is better to include a discussion section.
Reviewer 3 Report
The manuscript by Wagle et al. describes eddy covariance measurements and remote sensing observations for southern tallgrass prairies for non-growing seasons. As changes in growing season lengths and shoulder season environmental parameters like rainfall and temperature are changing due to climate change, I believe that this study is a valuable contribution to the scientific literature. However, before I can recommend this manuscript for publication it requires some issues to be addressed. I have two major concerns. 1) The manuscript needs further description on how gaps in remote sensing and eddy covariance data were handled, specifically for describing cumulative and binned values. Furthermore, the authors need to add an uncertainty analysis. Uncertainty of eddy covariance measurements can be very high during the non-growing season. To understand trends and differences between years and pastures, it is crucial to add an uncertainty analysis. 2) The manuscript is lacking a statistical analysis. Most results are descriptive, but it would be of value to better characterize these differences using statistics. For example, how were meteorological and phenological trends analyzed? It would be helpful to at least perform a correlation analysis for the observations, particularly as effects of rainfall on fluxes and vegetation greenness are described in the text. Eddy covariance fluxes are compared to vegetation greenness in the text, but it is questionable if these are real observed trends without proper uncertainty & statistical analyses. Further, remote sensing data needs to be described in more detail. The authors need to add some kind of measure of variability for these estimates for the different pastures, particularly as 500 meter and 250 meter resolution was used. It is unclear why two remote sensing products were used. More detailed comments are below:
Line 56: I’d like to note that this timescale varies, eddy covariance can be used on a variety of different time scales, ranging from minutes (using wavelet analysis) to hours.
Line 149: Please specify if gap-filling was performed using ustar filtering. I would also like to see an uncertainty analysis for the fluxes, as non-growing season fluxes usually have greater uncertainty intervals compared to growing season data. This should be fairly easy in REddyProc.
Line 155: Please describe data gaps for remote sensing data. This is of importance since you describe cumulative NDVI and EVI in your results. Furthermore, why did the authors only use Terra MODIS products? I am also wondering why the authors used both the 250 and 500 meter products.
Line 231: Please describe what “good rainfall” is in this context. I also don’t know what large sums of EVI and NDVI look like. Please be more specific.
Line 239: A cumulative value of GPP is shown here, but it is not clear from this section how GPP compares to vegetation greenness.
Line 254: I would argue that what you describe here is not a negligible effect of baling, but an interaction of environmental variables and management. Would this effect be negligible if rainfall rates did not increase?
Line 257: Replace bailing with baling
Line 278: I would recommend moving the grazing description to the methods section.
Line 307: This is why it would be crucial to understand flux uncertainty. I would argue that the difference in 24 g C m-2 to 26.5 g m-2 are minimal.
Line 324: This is of concern to me. How were averaging periods different? It is crucial to describe this in the methods in order to understand the results. I’m also wondering why section 2.2. describes gap-filling procedure, while the authors then used non-gapfilled data for their analysis.
Line 345: The sentence starting with “However, rainfall…” is confusing. I would recommend rewording.
Line 347: Please describe how rainfall can alter microbial activity in more detail
Line 370: For cumulative ET (and NEE and GPP in sections below) did you used gap-filled data? Please describe handling of flux data in more detail in your methods.
Table 1: Please add uncertainty of estimates
Line 410: How was the 30-year mean temperature calculated? Please describe in the methods section.
Line 413: Please add a description of how flux data binning was accomplished to your methods section.
Line 418: this is of concern to me. Why were different averaging periods used? And again, this needs to be described in the methods section.
Line 452: I would argue that without a proper statistical analysis it is difficult to defend arguments like this.
Line 471: Please describe in more detail how the presence of C3 species alters this relationship.
Round 2
Reviewer 2 Report
After the review by the authors, the paper has really improved. I recommend it for publication.
Reviewer 3 Report
The authors have addressed most if my comments and have added uncertainty estimates for their results, which was one of my main concerns. They have also added more descriptions on methods used, which is great. My recommendation is to accept.